# Suicidality in Adolescence: Insights from Self-Reports on Depression and Suicidal Tendencies

**DOI:** 10.3390/jcm14041106

**Published:** 2025-02-09

**Authors:** Marika Orlandi, Erica Casini, Diletta Cristina Pratile, Chiara Iussi, Cecilia Ghiazza, Renato Borgatti, Martina Maria Mensi

**Affiliations:** 1Department of Brain and Behavioral Sciences, University of Pavia, 27100 Pavia, Italy; 2Child Neurology and Psychiatry Unit, IRCCS Mondino Foundation, via Mondino 2, 27100 Pavia, Italy

**Keywords:** suicide, suicide ideation, suicide attempt, adolescence, self-report, prevention, depression

## Abstract

**Background and Objectives**. Suicide represents a primary global health concern, particularly among young individuals aged 15 to 29. Clinicians are actively engaged in efforts to prevent suicide and implement timely interventions. This study aimed to evaluate the effectiveness of self-reported measures in differentiating between adolescents exhibiting suicidal ideation (SI) only and those at risk or with a previous history of suicide attempts (SA). **Methods**. Seventy-eight adolescent patients (mean age: 15.53 ± 1.49) were classified into two groups using the Columbia Suicide Severity Rating Scale (C-SSRS). Forty-five patients presented with SI but lacked a prior history of SA, while 33 adolescents had a documented history of either concrete or interrupted SA. Notably, all participants in the SA group also reported SI. Participants completed the Multi-Attitude Suicide Tendency Scale (MAST) and the Beck Depression Inventory-Short Form (BDI-SF) to assess protective and risk factors associated with suicidality, as well as perceived depression. **Results**. Attraction toward life (AL) exhibited a negative correlation with perceived depression in both groups, whereas attraction toward death (AD) was positively correlated with depression in the SA group. In the SI group, scores for repulsion by life (RL) demonstrated a positive correlation with depression. Furthermore, RL scores were significantly higher in the SA group. ROC analysis revealed good accuracy for both assessment tools in differentiating the two groups. **Conclusions**. The BDI-SF and MAST are effective instruments for identifying adolescents at risk for suicide and implementing tailored preventive and therapeutic interventions. The user-friendly nature and adaptability make those self-report measures useful in various settings, allowing administration without clinician involvement.

## 1. Introduction

In 2024, suicide was the third leading cause of death worldwide among individuals aged 15 to 29 years [1], and the age at which suicidal ideation (SI) or suicide attempts (SA) occur is dropping alarmingly. In fact, recent evidence indicates that children aged 6 to 10 are already seeking mental health support for suicidality-related concerns [2]. Suicidal thoughts and behaviours arise from a complex interplay of social, psychological, sociocultural, and environmental factors [3,4]. Social factors such as peer rejection, bullying, and academic stress significantly contribute to adolescent suicidality [5,6]. Additionally, exposure to familial conflict, abuse, neglect, a family history of suicidal behaviours, substance abuse, or financial issues further amplifies suicide risk [7,8]. From a psychological perspective, adolescents with poor emotional regulation, impulsivity, and perceived burdensomeness face an increased risk for SI and SA [3,4]. Sociocultural factors, including stigma surrounding mental health, feelings of guilt or shame, and restrictive cultural norms regarding emotional expression, may further elevate suicide risk [9]. Moreover, environmental factors, such as limited access to health care, unsupportive school environments, and exposure to disasters, including hurricanes, earthquakes, accidental fires, armed conflicts, or pandemics, contribute to heightened suicidality risk [10].

The COVID-19 pandemic has profoundly impacted mental health, particularly among young individuals, exacerbating the prevalence of psychiatric disorders and symptoms such as anxiety, depression, post-traumatic stress disorder (PTSD), eating disorders, sleep disturbances, substance use disorders, self-harm, SI, and SA [11,12,13,14].

Consequently, clinicians must refine assessment methodologies and enhance preventive and therapeutic interventions. In fact, adolescents often engage in SA as a cry for help, as verbalising distress may be difficult for them [15]. This developmental stage is characterised by frequent crises, emotional maladjustments, impulsivity, and a heightened sense of urgency [16]. Some adolescents may perceive suicide as an escape from personal problems, a way to alleviate psychological distress or shame, or even a means of resolving social or familial conflicts [17,18]. Others may perceive it as a revenge [16]. Survivors of SA frequently encounter stigma and are at an increased risk of future attempts [19,20].

Given that adolescence is a critical period for the early detection, prevention, and intervention of suicidality [21], those risk factors must be addressed. Despite extensive efforts by researchers and clinicians to develop effective suicide prevention strategies, significant gaps remain in the literature [22,23]. Studies have identified inconsistencies in self-reported assessments of SI and behaviours, mainly due to factors such as recall bias, stigma, or differences in question phrasing (e.g., structured interviews vs. self-reports) and time points [24]. To overcome these limitations, previous research has explored the use of brief, structured questionnaires designed to identify key characteristics distinguishing individuals with SI from those with a history of SA, thereby enhancing risk identification and prevention efforts [25]. Additionally, self-report measures have been employed to examine the unique aspects of depression among suicidal adolescents [26]. While depressive disorders are not always present in individuals exhibiting suicidal behaviours, depression remains a significant risk factor for adolescent suicide [27,28].

Given the urgency of early detection, researchers have also considered involving additional caregivers in the screening process. Studies have sought to identify depressive symptoms and other relevant characteristics using tools that do not require clinician supervision [29]. For instance, prior research employed the Multi-Attitude Suicide Tendency Scale (MAST) [30] to investigate correlations between personality traits and suicidal behaviour [31], the presentation of depression in adolescents at risk for suicide [26], and the distinguishing characteristics of SA compared to those experiencing SI only [25].

This study aims to evaluate the efficacy of two brief, cost-effective self-report measures in differentiating between adolescents who experience SI exclusively and those who have a history of SA. We hypothesise that distinct differences will emerge between these two groups and that specific relationships among the questionnaire subscales will help identify adolescents at heightened risk for SA.

## 2. Materials and Methods

### 2.1. Study Design

This cross-sectional study was conducted following the REporting of studies Conducted using Observational Routinely collected health Data (RECORD) guidelines (Appendix A). The study protocol was approved by the Ethics Committee of Policlinico San Matteo in Pavia, Italy (P-20200055757), and adhered to the principles of the Declaration of Helsinki and its subsequent amendments [32]. Patients and their caregivers provided written informed consent to participate in the study, with the option to withdraw without justification. All data were pseudonymised and are available in the Zenodo repository [33].

### 2.2. Participants

The study enrolled 165 help-seeking male and female patients aged 12 to 18 years who were either outpatients or inpatients at the Child Neurology and Psychiatry Unit of the National Neurological Institute IRCSS Mondino Foundation in Pavia, Italy, between May 2020 and October 2022. A total of 87 patients were excluded based on the following criteria: intellectual disability (IQ ≤ 70), insufficient comprehension of the Italian language by the patients or their caregivers, absence of SI or SA, and lack of consent to participate in the study. The final study population is illustrated in Figure 1.

Patients were categorised into two groups based on their responses to the Columbia-Suicide Severity Rating Scale (C-SSRS) [34], a semi-structured interview designed to assess the frequency and severity of SI, as well as suicidal behaviour and SA.

The first group comprised patients who exhibited SI but did not provide affirmative responses to any of the suicidal behaviour items in the C-SSRS. Within this group, 53.33% reported a “passive” SI, with an intensity score ranging from 1 (“Wish to be Dead”) to 2 (“Non-Specific Active Suicidal Thoughts”), indicating a general desire to die without intent to engage in self-harm. The remaining 46.67% exhibited “active” SI, scoring between 3 (“Active Suicidal Ideation with Any Methods—Not Plan—without Intent to Act”) and 5 (“Active Suicidal Ideation with Specific Plan and Intent”) on the C-SSRS. “Active” SI denotes a persistent and specific contemplation of self-harm with an anticipated fatal outcome. Only 13.33% of the SI group reported an SI intensity of 5.

The second group comprised patients with a documented history of SA, including either concrete or interrupted attempts. These individuals were assessed using the suicidal behaviour items of the C-SSRS. They reported the ingestion of massive dosages of medications or toxic substances (such as bleach) and defenestration as primary methods to attempt their lives. Severe self-inflicted injuries resulting in severe blood loss, as well as hanging or the use of firearms, were rarely reported. Notably, all patients in the SA group also experienced SI. Specifically, 21.21% of the SA group exhibited “passive” SI (score 1 to 2 on the C-SSRS), while 78.79% reported “active” SI (score 3 to 5), with 48.49% reaching the highest SI severity score of 5.

### 2.3. Instruments

A child neuropsychiatrist collected sociodemographic and anamnestic data, including socioeconomic status (SES) [35]. SES levels were categorised as follows: low (8–19), middle-low (20–29), middle (30–39), middle-high (40–54), and high (55–66). We assessed social relationships according to four categories: social withdrawal (the participant is unable to function socially or maintain interpersonal relationships), poor social relationships (the participant may have some meaningful interpersonal relationships with peers but struggles with conflict resolution and developing or maintaining age-appropriate intimate relationships), or adequate social relationships (the participant engages in a wide range of social and interpersonal activities and exhibits age-appropriate involvement in intimate relationships). Moreover, academic performance was categorised as excellent (As), good (Bs), sufficient (Cs), poor (Ds and Fs), or school withdrawal (the participant is not attending school).

To exclude individuals with intellectual disabilities, we administered the Wechsler Intelligence Scale for Children-Fourth Edition (WISC-IV) [36], designed for children and adolescents aged 6 years to 16 years and 11 months, or the Wechsler Adult Intelligence Scale-Fourth Edition (WAIS-IV) [37], designed for adolescents and adults aged 16 years to 90 years and 11 months. Given the age overlap at 16 years, the choice between the WISC-IV and WAIS-IV often depends on the context, the individual’s cognitive development, and the assessment goals. For this study, to reduce biases, we administered the WAIS-IV to all adolescents from 16 years old.

A trained clinician or psychologist conducted the standardised clinical interview Kiddie Schedule for Affective Disorders and Schizophrenia-Present and Lifetime Version (K-SADS-PL) for DSM-5 [38,39] for the participants and/or their parents or guardians in separate sessions to confirm the psychiatric diagnoses and comorbidities. This semi-structured diagnostic interview assesses both lifetime and current psychopathological disorders in children and adolescents according to DSM-5 criteria, including mood disorders, eating disorders, obsessive-compulsive disorder (OCD), post-traumatic stress disorder (PTSD), mania, autism spectrum disorders, and substance use disorders. All diagnoses were established following DSM-5 criteria [40] and corroborated via the K-SADS-PL.

To assess the characterisation of SI, we employed:The Columbia Suicide Severity Rating Scale (C-SSRS) [34].

This semi-structured interview evaluates the severity and intensity of SI using a five-point scale from a generic desire to die to active suicidal ideation with a specific plan and intention. A score of 1 corresponds to the wish to be dead, 2 represents non-specific active suicidal thoughts, 3 denotes active SI with any method (without a specific plan) and without intent to act, 4 signifies active SI with some intent to act but lacking a specific plan, and 5 reflects active SI with a detailed plan and purpose. Additionally, this scale assesses behaviours that may indicate an individual’s intention to commit suicide and emphasises previous SA. The C-SSRS examines the occurrence of actual, interrupted, and aborted suicide attempts throughout an individual’s lifetime (using a dichotomous response format), the frequency of each type of attempt, preparatory actions (also using a dichotomous response), and the actual and potential lethality associated with suicidal behaviours in the context of concrete attempts.

The Multi-Attitude Suicide Tendency Scale (MAST) [30].

This 30-item self-report questionnaire evaluates suicide risk and protective factors using a Likert scale from 1 (strongly disagree) to 5 (strongly agree). It comprises four subscales investigating subjective attitudes as mediators of suicidality without differentiating between SI and SA.

(1)Repulsion by Death (RD): RD may be significantly elevated even among individuals exhibiting a pronounced inclination towards self-destructive behaviours. This phenomenon arises from the recognition that death is an unavoidable conclusion to life. RD serves as a substantial deterrent against self-destructive tendencies, potentially influenced by the apprehension of severe repercussions following death. A score equal to or lower than 3 indicates risk, while a higher score is considered a protective factor. The Cronbach’s alpha coefficient for this measure was 0.92.(2)Attraction toward Life (AL): Attraction toward Life (AL) is significantly influenced by an individual’s sense of security within interpersonal relationships, including those with family and friends, as well as romantic relationships and self-esteem. This construct generally serves as a deterrent to self-destructive behaviours. A score equal to or lower than 3 indicates risk, while a higher score is considered a protective factor. The reliability of this measure, as indicated by Cronbach’s alpha, was 0.85.(3)Repulsion by Life (RL): RL encompasses experiences of suffering, including the inability to resolve feelings of guilt, the loss of loved ones, experiences of abuse, tendencies toward self-destructive behaviour, and identification with depressed parents. A score of 3 or higher indicates a risk, while a score below 3 is a protective factor. The reliability of this measure, as indicated by Cronbach’s alpha, was 0.79.(4)Attraction to Death (AD): AD arises from a distorted perception of death as reversible and as a potential escape from life’s challenges. Death is frequently idealised during adolescence as a mystical union with the universe. For individuals who have experienced the loss of a loved one, this may represent a desire to reunite with the deceased. A score of 3 or higher on the AD scale is associated with an increased risk, while a lower score is a protective factor. The scale’s reliability, as measured by Cronbach’s alpha, was 0.79.

The reliability of the MAST in distinguishing risk and protective factors in adolescent suicidality has been established in prior research [41,42,43].

To evaluate the severity of patients’ perceived depression, we administered:The Beck Depression Inventory-Short Form (BDI-SF) [44]

This 13-item self-report questionnaire employs a 4-point Likert scale (0–3). Higher scores indicate greater severity of depressive symptoms, with the following classifications: scores of 5–7 denote mild symptoms, scores of 8–15 indicate moderate symptoms, and scores of 16 or higher are categorised as severe. The items assess various dimensions of depressive symptoms, including sadness, pessimism, feelings of failure, dissatisfaction, guilt, self-esteem, self-harm, social withdrawal, indecision, self-image, difficulties in work performance, fatigue, and appetite disturbances. The reliability of the questionnaire is supported by a Cronbach’s alpha coefficient of 0.91.

### 2.4. Analysis

The analyses were conducted utilising JASP (Version 0.19.1) [45]. Descriptive statistics were computed for sociodemographic variables. The internal consistency of the questionnaires was assessed using Cronbach’s alpha. A paired samples *t*-test was employed to investigate differences between the two groups based on categorical variables. The Mann–Whitney U test assessed differences between the two groups concerning the MAST and BDI-SF scales. This non-parametric test was chosen due to the non-normal distribution of the data, as it does not assume normality and is appropriate for comparing independent samples. Normality was assessed using the Shapiro–Wilk test, confirming that the data did not follow normal distribution. The relationship between the BDI-SF and the MAST scores was examined using Spearman’s rank correlation. Bonferroni correction was applied to reduce the probability of type I errors due to multiple testing. Significance levels are reported as *p*-values, with values below 0.05 considered statistically significant. Finally, receiver operating characteristic (ROC) analyses were conducted with R [46] to quantify the accuracy of these measures in discriminating between the two groups.

## 3. Results

### 3.1. Descriptive Analyses

We enrolled 78 adolescents, categorising them into two groups based on the C-SSRS criteria. The SI group comprised 45 participants, while the SA group included 33 participants. Descriptive analyses indicated that the groups were homogeneous in age, ethnicity, family status (presence of separated or divorced parents), and SES. However, a notable difference emerged in gender distribution, with a higher proportion of females in the SA group. Moreover, differences were observed in active and passive SI, with the SA group exhibiting a higher prevalence of active SI (Table 1).

### 3.2. MAST Differences Between the SI and SA Groups

After applying the Bonferroni correction, the analyses identified a significant difference between the two groups in the RL subscale of the MAST, as shown in Table 2.

### 3.3. Differences Between the SI and SA Groups Concerning BDI-SF

The Mann–Whitney U test revealed a statistically significant difference in depression levels between the two groups (U = 404.00, *p* = 0.003; Adj *p* = 0.015). Specifically, the SA group exhibited higher levels of depression (M_SA_ = 23.97, SD = 9.69; min 0, max 35; M rank = 49.97) compared to the SI group (M_SI_ = 16.07, SD = 11.12; min 0, max 37; M rank = 31.68). The area under the curve (AUC) for the BDI-SF was 0.706, indicating good accuracy (Figure 2).

### 3.4. Comparison Between the MAST and BDI-SF

Within both groups, higher levels of AL were associated with lower levels of depression. Conversely, elevated AD correlated with increased depression levels, but only in the SA group. Notably, in the SI group, higher RL scores were linked to increased BDI-SF scores (Table 3).

Finally, ROC analyses evaluating the discriminative power of the MAST and BDI-SF in distinguishing between the two groups yielded an AUC of 0.727, which is considered good (Figure 3).

## 4. Discussion

This study aimed to evaluate the effectiveness of two brief and cost-less self-report measures in distinguishing between adolescents exhibiting SI and those at risk or with a previous history of SA. Building upon prior research [25], we first compared MAST subscales across the two groups. Results indicated significant differences in the RL subscale, with higher scores in the SA group, consistent with the findings by Maggiolini et al. [47]. Participants exhibited reduced coping capacity, unsolved guilt, and limited bodily awareness, in line with prior studies highlighting a prevalence of internalising disorders, such as depressive disorders [27,28]. Additionally, they expressed greater social withdrawal [25,47], a factor frequently associated with loneliness, non-suicidal self-injury (NSSI), SI, SA, and deliberate self-harm, with or without suicidal intent [48]. Furthermore, the literature underscores that feelings of loneliness, helplessness, and hopelessness are strongly linked to social withdrawal, depression, and SI or SA [48,49,50]. Although these constructs were not directly assessed in the present study, they represent critical areas for further research to enhance the understanding of suicide risk factors in adolescents.

The existing literature highlights depression as a significant risk factor in the aetiology of suicidal behaviour among adolescents [27,28]. Individuals experiencing depressive symptoms often express a desire to die or engage in suicidal behaviour, perceiving them as a potential solution to their problems [51]. A previous study [50] identified severe depression as a key predictor of SI in both adolescents and adults.

To further examine the role of depression, we administered the BDI-SF. The results indicated that the SA group reported higher levels of depressive symptoms compared to the SI group. This finding aligns with previous studies [25] and reinforces depression as a specific risk factor for SA. Consequently, the BDI-SF serves a predictive function, facilitating the early identification of at-risk individuals. Timely access to appropriate psychological assessment and treatment may alleviate patients’ depressive symptoms and reduce suicide risk.

The comparative analysis of BDI-SF and MAST scores revealed a negative correlation between AL scores and perceived depression in both groups. This finding is consistent with Maggiolini et al., who linked low AL scores to internalising symptoms [47]. AL reflects an individual’s positive outlook on life, signifying the pleasantness of their existence and enhancing self-affirmation and interpersonal relationships [52]. Thus, high levels of AL could be considered a protective factor against depression and suicidality.

Furthermore, we interestingly observed that higher AD scores correlated with increased perceived depression levels, but only in the SA group. AD represents a distorted perception of death as reversible and as an escape from life’s challenges or a way to reunite with someone deceased. This is in line with previous studies stating that an increased risk of SA accompanies a distorted view of reality and the presence of psychotic features [53].

Moreover, we observed a positive correlation between high RL levels and BDI-SF scores in the SI group, suggesting that higher RL levels are associated with increased depressive symptoms. This could be explained because RL encapsulates profound experiences of trauma, guilt, loss, and self-destructive tendencies. This difference could also be related to the differences in active and passive SI expressed by the two groups. A previous study highlighted that patients with SA show more intense SI severity according to the C-SSRS [25]. This supports the importance of assessing intent and planning, as these factors significantly increase the risk of SA.

Furthermore, a comparison between the two samples revealed differences in the diagnosis of depressive disorders. Despite the higher presence of perceived depression in the SA group, according to BDI-SF, individuals with SI were more frequently diagnosed with depressive disorders, albeit with milder symptoms. Additionally, personality disorder (PD) features were more prevalent in the SA group, particularly borderline PD traits, such as impulsivity and anxious lability, which have emerged as significant markers for suicidality and SA relapse [54]. This finding is consistent with prior research indicating that individuals with SA may exhibit higher levels of impulsivity than those with SI, particularly under negative emotional states [55]. Another study found that impulsive reactivity to emotional stimuli is strongly linked to SA, even when controlling for psychiatric diagnoses and symptoms [56]. While our findings align with research suggesting that patients with SA exhibit more PD-related symptoms [25], it is essential to note that diagnosing PD at a young age is not always considered accurate. Clinicians should closely monitor adolescents exhibiting high-risk PD traits over time to track symptom trajectories. In light of that, recent research involving adolescents suggests that early onset of symptoms, externalising behaviours, and caregiver-reported internalising symptoms may predispose adolescents to fully developed PDs in adulthood [57]. Our findings support the notion that suicidality is transdiagnostic, cutting across multiple psychiatric conditions [58].

The ROC analysis provided further evidence that the BDI-SF and MAST demonstrated good accuracy in distinguishing between adolescents with SI and those with a history of SA. These results suggest that both tools can serve as reliable initial screening measures, particularly in non-clinical settings where quick, cost-effective assessments are needed. However, depression manifests heterogeneously, making its diagnosis inherently challenging [59]. Suicidality represents a complex construct that includes diverse symptoms and varying degrees of severity [60].

Nevertheless, differentiating between adolescents with SI and those with SA remains complex [61]. Additionally, assessing at-risk adolescents is further complicated by the challenge of identifying factors that facilitate the transition from ideation to attempt. This understanding is crucial for developing timely and effective interventions and preventive strategies.

Our findings suggest that self-reported measures may be valuable early screening tools in non-clinical settings, such as schools. However, effective screening requires a multimethod approach, incorporating multiple instruments and perspectives. Unfortunately, timely access to mental health services remains inconsistent.

The study presents some limitations. First, the sample demonstrates gender heterogeneity, as the SA group comprises primarily female participants. This reflects the literature stating that female adolescents present a higher risk for SA and males for suicide death [62]. Moreover, research shows differences in help-seeking tendencies, with young males asking for help from those they trust (e.g., friends and parents) and adopting self-reliance as the preferred strategy to cope with mental issues. At the same time, females seem to have more confidence in mental health professionals [63]. Second, both groups consist solely of individuals diagnosed with severe neuropsychiatric disorders and do not include a healthy control group. Furthermore, the participants were self-selected; they consented to participate in the study from a larger cohort admitted to our institution. Moreover, the study’s cross-sectional design, even if inexpensive and quick to conduct, does not allow for the determination of causality. A final limitation is associated with the use of two self-administered questionnaires, which may result in socially desirable responses, inaccurate self-assessment, or an exaggeration of psychopathological conditions.

## 5. Conclusions

The study underscores the potential of the MAST and BDI-SF as practical self-report tools for differentiating adolescents with SI from those with or at risk of SA. Their ease of use makes them valuable for early screening, particularly in non-clinical settings. However, given the potential discrepancies between clinician evaluation and self-reported symptoms, it is crucial to integrate various assessment methods. Clinicians should also actively involve important caregivers to develop tailored prevention strategies. Implementing evidence-based screening protocols and multidisciplinary prevention programs may help reduce suicide risk factors and enhance protective ones. Given that suicide is transdiagnostic and recognised as a leading cause of adolescent mortality worldwide, early detection through screening and intervention is essential [64].

The study’s strengths are primarily attributed to the user-friendly design of the questionnaires. Surveying varied contexts would yield valuable insights, as suggested by prior research involving the general population [11,12]. The implementation of written questionnaires facilitates expression for individuals reluctant to verbalise their feelings. Additionally, the reliability of C-SSRS further enhances the robustness of the study [65]. Nevertheless, considering the complex nature of suicide risk factors, future research should expand these preliminary findings by investigating a larger, more heterogeneous sample, including a control group. Future studies could also incorporate clinical observations and explore additional risk factors such as loneliness, helplessness, and hopelessness.

## Figures and Tables

**Figure 1 jcm-14-01106-f001:**
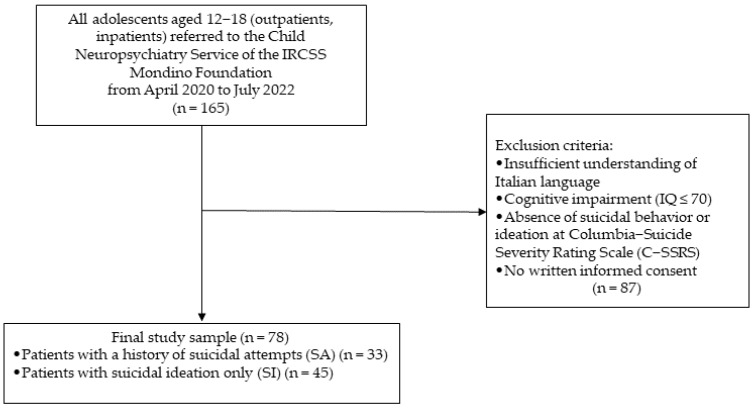
Study population [author’s own processing].

**Figure 2 jcm-14-01106-f002:**
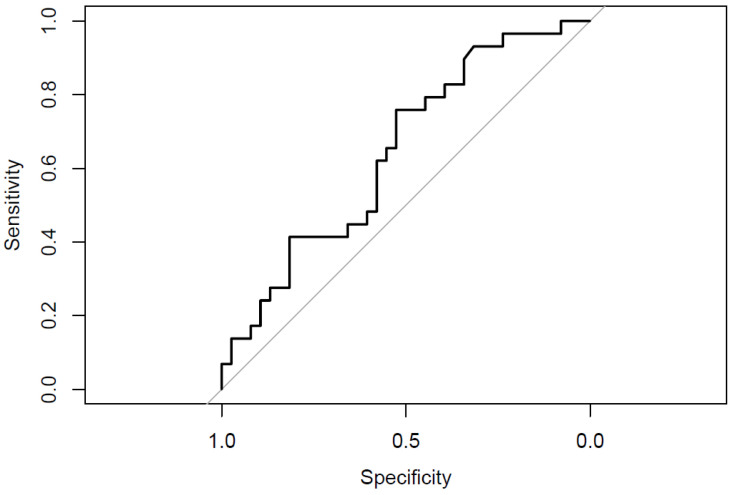
ROC curve illustrating the ability of BDI-SF to differentiate between SI and SA groups [author’s own processing].

**Figure 3 jcm-14-01106-f003:**
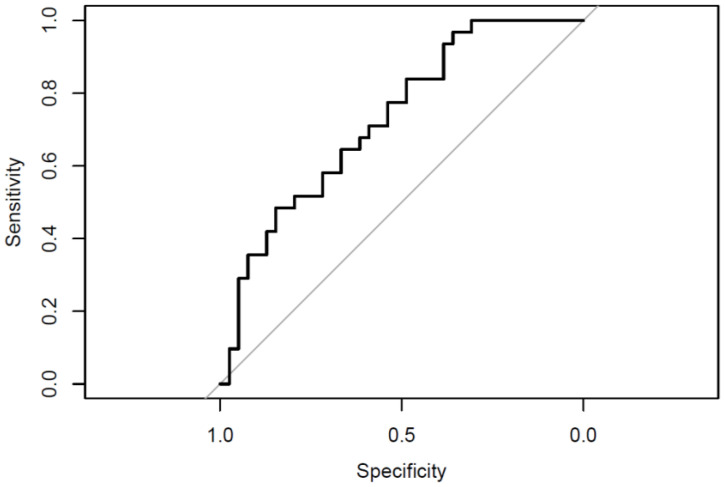
ROC curve illustrating the ability of BDI-SF and MAST to differentiate between SI and SA groups [author’s own processing].

**Table 1 jcm-14-01106-t001:** Sociodemographic data and diagnoses according to DSM-5 criteria.

	Total (n = 78) n (%)	SI (n = 45) n (%)	SA (n = 33) n (%)	** * χ * ** ^2^	df	*p*
Age (Mean ± SD)	15.53 ± 1.49	15.39 ± 1.66	15.66 ± 1.66	0.81	76	0.209
Gender (female)	69 (88.46)	37 (82.22)	32 (96.97)	4.06	1	0.044
Ethnicity				2.47	4	0.649
Caucasian	67 (85.90)	37 (82.22)	30 (90.91)			
Asian	1 (1.28)	1 (2.22)	0 (0.00)			
African	5 (6.41)	3 (6.67)	2 (6.06)			
Latina	3 (3.84)	2 (4.44)	1 (3.03)			
Mixed	2 (2.56)	2 (4.44)	0 (0.00)			
Separate/divorced parents	26 (33.33)	13 (28.89)	13 (39.39)	0.875	1	0.350
SES ^a^				2.33	4	0.675
Low	11 (14.10)	8 (17.78)	3 (9.09)			
Medium-low	10 (12.82)	5 (11.11)	5 (15.15)			
Medium	16 (20.51)	8 (17.78)	8 (24.24)			
Medium-high	17 (21.80)	8 (17.78)	9 (27.27)			
High	7 (8.97)	3 (6.67)	4 (12.12)			
School performances ^b^				1.180	4	0.881
Poor	14 (17.95)	7 (15.56)	7 (21.21)			
Sufficient	18 (23.08)	9 (20.00)	9 (27.27)			
Good	26 (33.28)	16 (35.56)	10 (30.30)			
Excellent	14 (17.95)	9 (20.00)	5 (15.15)			
School withdrawal	5 (6.41)	3 (6.67)	2 (6.06)			
Social relations ^c^				3.689	2	0.158
Social withdrawal	10 (12.82)	3 (6.67)	7 (21.21)			
Poor	44 (56.41)	26 (57.78)	18 (54.55)			
Adequate	23 (29.49)	15 (33.33)	8 (24.24)			
Risky behaviours ^d^	49 (62.82)	21 (46.67)	28 (84.85)	18.235	6	0.006
Active SI	47 (60.26)	21 (46.67)	26 (78.79)	8.202	1	0.004
DSM-5 diagnosis						
Learning disability	2 (2.56)	1 (2.22)	1 (3.03)	2.098	2	0.350
Psychotic disorders ^e^	17 (21.79)	6 (13.32)	11 (33.33)	5.625	3	0.131
Bipolar and related	3 (3.84)	2 (4.44)	1 (3.03)	2.834	2	0.242
Depression	46 (58.88)	24 (53.28)	22 (66.66)	10.548	4	0.032
Disruptive mood dysregulation	2 (2.56)	2 (4.44)	0 (0)			
MDD	22 (28.20)	10 (22.2)	12 (36.36)			
Persistent depressive disorder	7 (8.97)	1 (2.22)	6 (18.18)			
Unspecified depressive disorder	15 (19.23)	11 (24.42)	4 (12.12)			
Anxiety	25 (32.00)	15 (33.3)	10 (30.3)	4.962	5	0.420
Specific phobia	1 (1.28)	1 (2.22)	0 (0)			
Social anxiety disorder	3 (3.84)	2 (4.44)	1 (3.03)			
Panic disorder	3 (3.84)	1 (2.22)	2 (6.06)			
GAD	8 (10.25)	3 (6.66)	5 (15.15)			
Unspecified anxiety disorder	10 (12.82)	8 (17.77)	2 (6.06)			
OCD	3 (3.84)	2 (4.44)	1 (3.03)	0.103	1	0.748
Eating disorders	20 (25.6)	14 (31.08)	6 (18.18)	6.859	4	0.144
Personality disorders (PD)	26 (33.28)	9 (19.98)	17 (51.51)	9.194	3	0.027
High risk for PD(s) ^f^	23 (29.49)	8 (17.78)	15 (45.46)			
Borderline PD	3 (3.84)	1 (2.22)	2 (6.06)			
Substance use disorders	2 (2.56)	1 (2.22)	1 (3.03)	0.050	1	0.823

Legend. Significance: *p* < 0.05; ^a^ SES: low (8–19), middle-low (20–29), middle (30–39), middle-high (40–54), high (55–66); ^b^ school performances: poor (Ds and Fs), sufficient (Cs), good (Bs), excellent (As), school withdrawal; ^c^ social relations: social withdrawal (the participant is unable to function socially or maintain interpersonal relationships), poor social relationships (the participant may have some meaningful interpersonal relationships with peers but struggles with conflict resolution and developing or maintaining age-appropriate intimate relationships), or adequate social relationships (the participant engages in a wide range of social and interpersonal activities and exhibits age-appropriate involvement in intimate relationships); ^d^ risky behaviours: non-suicidal self-injury (cutting or other), substance use, attempted suicide, non-suicidal self-injury and substance use, non-suicidal self-injury and attempted suicide, substance use and attempted suicide, all the previous; ^e^ psychotic disorders: brief psychotic disorder, unspecified schizophrenia spectrum and other psychotic disorder, clinical high risk for psychosis (CHR-P); bipolar and related: cyclothymic disorder, Unspecified Bipolar and Related Disorder; ^f^ high risk for PD(s): since diagnosing a PD in children and adolescents is not always appropriate, minors who exhibit characteristics of subthreshold PD but do not fully meet the criteria of the DSM-5 are considered at high risk for PD(s); GAD: generalised anxiety disorder; MDD: major depressive disorder; OCD: obsessive-compulsive disorder; ODD: oppositive defiant disorder.

**Table 2 jcm-14-01106-t002:** MAST differences between the two groups.

	SI		SA				
	M	SD	M	SD	t	*p*	Adj *p*
MAST-AL	2.83	0.85	2.45	0.68	2.030	0.023	0.115
MAST-RL	3.28	0.90	3.75	0.55	2.691	0.005	0.025
MAST-AD	3.16	0.82	3.47	0.57	1.896	0.031	0.155
MAST-RD	2.46	1.02	2.12	0.81	1.523	0.066	0.33

Legend: Significance: *p* < 0.05. AL: attraction toward life; RL: repulsion by life; AD: attraction toward death; RD: repulsion by death.

**Table 3 jcm-14-01106-t003:** Correlations between BDI-SF and MAST.

		BDI-SF				
	SI			SA		
	r	*p*	Adj *p*	r	*p*	Adj *p*
MAST AL	−0.633	<0.001	<0.001	−0.567	0.001	0.005
MAST RL	0.638	<0.001	<0.001	0.242	0.198	0.99
MAST AD	0.365	0.024	0.12	0.557	0.001	0.005
MAST RD	0.056	0.737	3.685	−0.332	0.073	0.365

Legend: Significance: *p* < 0.05. AL: attraction toward life; RL: repulsion by life; AD: attraction toward death; RD: repulsion by death.

## Data Availability

The data underlying this article are available upon reasonable request in Zenodo (10.5281/zenodo.10355718) [33].

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
