# Peer review of "Suicidality in Adolescence: Insights from Self-Reports on Depression and Suicidal Tendencies"

_jcm, 2025, doi:10.3390/jcm14041106_

Round 1

Reviewer 1 Report (Previous Reviewer 3)

Comments and Suggestions for Authors

I would like to thank the authors for their response to previous reviews and comments from the reviewers. The revised version of the manuscript has addressed my original concerns.

Author Response

Comment: I would like to thank the authors for their response to previous reviews and comments from the reviewers. The revised version of the manuscript has addressed my original concerns.

Response: We appreciate the reviewer's time to read our work and the valuable comments and support.

Reviewer 2 Report (New Reviewer)

Comments and Suggestions for Authors

Many thanks to have the  opportunity to revise this manuscript on evaluate the efficacy of two brief, cost-effective self-reports in differentiating between adolescents who exhibit SI exclusively and those who have a history of SA.

My comments and suggestions: 

- in the introduction, the authors should explain better the potential clinical, psychological and environmental variables that could lead to suicide in adolescence.

- PD can not be diagnosed before 18 years old and use "structuring PD" is not correct because they can not know the illness trajectories of this clinical dimension. It is the major limitation. I believe that the authors should change the psychiatric diagnosis or delete from the total sample. 

- female gender is over-represented, please discuss it in discussion and limitation section.  

- I suggest to delete the following part: "We selected the Beck Depression Inventory-Short Form (BDI-SF) to assess depressive symptoms, in conjunction with the MAST to examine attitudes and suicidal tendencies. Furthermore, we sought to identify the subscales that most effectively 74 indicate belonging to one group instead of the other, thereby elucidating protective and risk factors."

- are you sure that "learning disability diagnosis"is not in contrast with the inclusion criteria and shoul  d be excluded in the analysis? 

- "Dysthymia" in not considered in DSM 5, it is called in other way. 

- in material and methods, the authors should explain better the classification of the following variables"socioeconomic status" and "school performances" and "social relations" 

- Please add the number of ethical approvation. 

- Substitute the reference of "Declaraton of helsinki, because it has been modified in 2000. 

- The ROC analyses and curve are missing. Please insert as a figure. 

- Another limitation is the cross-sectional study design. 

- Several references could be add to describe better this topic as: 10.2147/PRBM.S279829, 10.23750/abm.v91i3-S.9417, 10.1016/j.jpsychores.2022.110788,

- In the ICD 11, also hopelessness is considered a depressive symptoms and the literature is full of articles on hopelessness, depression and suicide. Please add a comment in discussion, analysing also loneliness and helplessness. 

Author Response

Comment: Many thanks to have the opportunity to revise this manuscript on evaluate the efficacy of two brief, cost-effective self-reports in differentiating between adolescents who exhibit SI exclusively and those who have a history of SA.

Response: Thank you for the precious comments. We replied point-by-point.

Comment: My comments and suggestions:

- in the introduction, the authors should explain better the potential clinical, psychological and environmental variables that could lead to suicide in adolescence.

Response: We addressed the valuable reviewer’s comment explaining in-depth social, psychological, sociocultural, and environmental factors that contribute to the emergence of suicidal thoughts and behaviours in the Introduction. (lines 45-55)

Comment: - PD can not be diagnosed before 18 years old and use "structuring PD" is not correct because they can not know the illness trajectories of this clinical dimension. It is the major limitation. I believe that the authors should change the psychiatric diagnosis or delete from the total sample.

Response: We changed the term to make the manuscript more precise. We followed recent findings from another research stating that, although it is not possible to predict the developmental trajectories of a personality disorder from adolescence onward, one can speak of “adolescents at high risk for personality disorder(s)”.

Comment: - female gender is over-represented, please discuss it in discussion and limitation section. 

Response: We thank the reviewer for raising this point. We addressed the comment by discussing this issue further and adding appropriate references.

Comment: - I suggest to delete the following part: "We selected the Beck Depression Inventory-Short Form (BDI-SF) to assess depressive symptoms, in conjunction with the MAST to examine attitudes and suicidal tendencies. Furthermore, we sought to identify the subscales that most effectively 74 indicate belonging to one group instead of the other, thereby elucidating protective and risk factors."

Response: We addressed the reviewer’s comment and deleted that part.

Comment: - are you sure that "learning disability diagnosis"is not in contrast with the inclusion criteria and shoul  d be excluded in the analysis?

Response: We thank the reviewer for the comment. We specify that the patients with learning disability who participated in the study were 2 out of 78, as shown in Table 1. These patients had a mild learning disability, which did not affect their reading abilities or comprehension or compromise their participation in the study or the test results.

Comment: - "Dysthymia" in not considered in DSM 5, it is called in other way.

Response: We substituted the term “Dysthymia” with “Persistent depressive disorder”, even if the DSM-5 reported both “Persistent Depressive Disorder (Dysthymia)” 300.4 (F34.1)

Comment: - in material and methods, the authors should explain better the classification of the following variables"socioeconomic status" and "school performances" and "social relations"

Response: We thank the reviewer for the opportunity to deepen this point. We enlarged the classification reported in the Material and Methods section, at the beginning of Instruments section. We also specified that in Table 1. (lines 152-163 and 275-282)

Comment: - Please add the number of ethical approvation.

Response: We also reported the ethical approval number in the Design section, as requested.

Comment: - Substitute the reference of "Declaraton of helsinki, because it has been modified in 2000.

Response: We updated the reference.

Comment: - The ROC analyses and curve are missing. Please insert as a figure.

Response: We appreciated the reviewer’s suggestion. We addressed the comment by adding the appropriate figures (Figures 2 and 3).

Comment: - Another limitation is the cross-sectional study design.

Response: We thank the reviewer for raising this point. We updated the limitations section.

Comment: - Several references could be add to describe better this topic as: 10.2147/PRBM.S279829, 10.23750/abm.v91i3-S.9417, 10.1016/j.jpsychores.2022.110788,

Response: We thank the reviewer for the precious suggestion. We have read the recommended articles. Unfortunately, the first one represents a proposal for a study and does not refer to the adolescent population, and the other two refer to an adult population. We have explored the literature to identify studies that included adolescents, given the specific characteristics of this group, which in many aspects differ from the adult population. Compared to the works already published on the topic of Demoralization, this aspect is investigated by specific scales, such as the Demoralization Scale, which was not administered in our study and, to our knowledge, is validated in Italian only for adults. Similarly, the aspect of Meaning in Life (MiL) was not investigated in our study by specific instruments, as in other studies in the literature, and it is validated in Italian only for the adult population, too. Nevertheless, we revised the Introduction according to the reviewers’ suggestions to make it more straightforward.

Comment: - In the ICD 11, also hopelessness is considered a depressive symptoms and the literature is full of articles on hopelessness, depression and suicide. Please add a comment in discussion, analysing also loneliness and helplessness.

Response: Thank you for the insightful comment regarding including hopelessness, loneliness, and helplessness in the discussion. We have revised the manuscript to acknowledge the relevance of these constructs, as highlighted in the literature. While our study did not directly assess these aspects, we have added a comment in the discussion to reflect on their potential role in the context of depression and suicidality based on existing evidence. Specifically, we have introduced the following statement with appropriate references:

"Furthermore, the literature highlights that feelings of loneliness, helplessness, and hopelessness are frequently associated with social withdrawal, depression, and SI or SA [48–50]. Although these constructs were not directly assessed in the present study, they represent critical areas for further exploration to enhance the understanding of suicide risk factors in adolescents." (lines 343-347)

This addition allows us to address the reviewer’s observation while focusing on our study's specific aims and findings. We hope this revision aligns with the reviewer’s suggestion and enhances the discussion.

Reviewer 3 Report (New Reviewer)

Comments and Suggestions for Authors

1-)you can also mention the gap in the literature in the background section.

2-)in the abstract section, you should only mention the main conclusion of your study.

3-)you may consider making the title of your study clearer.

4-)you should mention the year (e.g., 2024, 2025).

Suicide ranks as the second leading cause of death worldwide among individuals aged 15 to 29 years

5-)you can mention the names of psychiatric disorders.

exacerbating the prevalence of psychiatric disorders during adolescence.

6-)you should explain it further:

Some research has highlighted inconsistencies in self-reports used to assess suicide risk [15]

7-)please correct the typo.

design. line 80.

8-)be careful about the punctuation.

line 81

9-)be sure the title is concise.

Study population flowchart [author’s own processing]. line 99

10-)you can give more information about it:

based on the age of the patient

11-)you can give example of two groups and explain further if possible.

istic (ROC) analyses were conducted to quantify the 214
accuracy of these self-reports in discriminating between the two groups

12-)you may consider mentioning the main results of your study in the discussion section first.

Author Response

Comment: 1-)you can also mention the gap in the literature in the background section.

Response: We thank the reviewer for the precious comment. We revised the introduction to clarify this point.

Comment: 2-)in the abstract section, you should only mention the main conclusion of your study.

Response: We thank the reviewer for the valuable suggestion. We extensively rewrote the abstract to make it more precise and compelling.

Comment: 3-)you may consider making the title of your study clearer.

Response: We appreciated the reviewer’s comment. We modified the title to better express the purpose of the study. The new title is “Suicidality in Adolescence: Insights from Self-Reports on Depression and Suicidal Tendencies”

Comment: 4-)you should mention the year (e.g., 2024, 2025).

Suicide ranks as the second leading cause of death worldwide among individuals aged 15 to 29 years

Response: We thank the reviewer for raising this issue. We updated the data in the Introduction and the reference.

Comment: 5-)you can mention the names of psychiatric disorders.

exacerbating the prevalence of psychiatric disorders during adolescence.

Response: We appreciated the reviewer’s comment. We deepened the Introduction by adding the names of psychiatric disorders and updating the references. (lines 58-61)

Comment: 6-)you should explain it further:

Some research has highlighted inconsistencies in self-reports used to assess suicide risk [15]

Response: Thank you very much for the suggestion. We better explained this sentence. The sentence has been modified as follows “Some research has highlighted significant discrepancies in self-reports used to assess suicidal ideation and behaviours depending on the assessment method used [citation]. Individuals provide inconsistent responses across different instruments (e.g., structured interviews, self-reports) and time points, primarily because of recall bias, stigma, or differences in question phrasing.” (lines 74-79)

Comment: 7-)please correct the typo.

design. line 80.

Response: We thank the reviewer. We corrected it.

Comment: 8-)be careful about the punctuation.

line 81

Response: We addressed the comment. We also checked the punctuation in the entire manuscript.

Comment: 9-)be sure the title is concise.

Study population flowchart [author’s own processing]. line 99

Response: We fixed it according to the reviewer’s suggestion and the journal's indications.

Comment: 10-)you can give more information about it:

based on the age of the patient

Response: We thank the reviewer for raising his point and allowing us to better explain this in the manuscript. We specified it in the “Instruments” section. The new paragraph states “To exclude individuals with intellectual disabilities, we administered the Wechsler Intelligence Scale for Children-Fourth Edition (WISC-IV) [citation], designed for children and adolescents aged 6 years to 16 years and 11 months, or the Wechsler Adult Intelligence Scale-Fourth Edition (WAIS-IV) [citation], designed for adolescents and adults aged 16 years to 90 years and 11 months. There is an overlap for 16-year-olds. Usually, the choice between the WISC-IV and WAIS-IV often depends on the context, the individual's cognitive development, and the assessment goals. For this study, to reduce biases, we administered the WAIS-IV to all adolescents from 16 years old”. (lines 164-172)

Comment: 11-)you can give example of two groups and explain further if possible.

istic (ROC) analyses were conducted to quantify the 214

accuracy of these self-reports in discriminating between the two groups

Response: We appreciated the reviewer’s comment. In the Results section, we added figures to explain ROC results better and further discussed them in the Discussion. (lines 404-407)

Comment: 12-)you may consider mentioning the main results of your study in the discussion section first.

Response: We thank the reviewer for the valuable suggestion. We extensively edited the Discussion section to address the comment.

Round 2

Reviewer 2 Report (New Reviewer)

Comments and Suggestions for Authors

The authors have addressed all my suggestions clearly. 

Comments on the Quality of English Language

I think that a language revision is needed

Author Response

Comment: I think that a language revision is needed

Response: We addressed the reviewer’s comment about having extensive English editing.

This manuscript is a resubmission of an earlier submission. The following is a list of the peer review reports and author responses from that submission.

Round 1

Reviewer 1 Report

Comments and Suggestions for Authors

There are very primary design problems in this study. The main problem is the fact that the study does ot have a calculation of the size of the sample and it is analysing too much factors with the same small number of individuals included. The analysis does not describe how the decisions to ues 𝟀2 or T Test  was madet. The Descriptive analyses extensevly described the values of 𝟀2. I is my opinion that such information belongs in the Table 1. It is important to describe what are MAST RL, RA, AL, AR in a legend. The comparison of MAST with all the other scales needs to be better explained and supported  according to the number needed to compare results. 

There is an excess of tables that are short and do not contain a great number oflittle  information. 

Comments on the Quality of English Language

There are some typos that need correction

Author Response

Comment 1: There are very primary design problems in this study. The main problem is the fact that the study does ot have a calculation of the size of the sample and it is analysing too much factors with the same small number of individuals included. The analysis does not describe how the decisions to ues ?2 or T Test  was madet. The Descriptive analyses extensevly described the values of ?2. I is my opinion that such information belongs in the Table 1. It is important to describe what are MAST RL, RA, AL, AR in a legend. The comparison of MAST with all the other scales needs to be better explained and supported  according to the number needed to compare results.

Response 1: We appreciated the reviewer’s comments and updated the manuscript accordingly. In section 2.4, we added the sample size calculation and explained the decision to use the chi-square or t-test and why we compared MAST and BDI-SF. Moreover, we moved the descriptive analyses to Table 1, enriching it. We also added a legend specifying MAST subscales RL, RD, AL, and AD acronyms. For an in-depth explanation of those subscales, we refer the reader to the Instruments section (2.3).

Comment 2: There is an excess of tables that are short and do not contain a great number oflittle  information.

Response 2: We thank the reviewer for this suggestion. We reduced the number of tables to make the manuscript clearer.

Comment 3: There are some typos that need correction.

Response 3: We reviewed the entire manuscript to correct the typos.

Reviewer 2 Report

Comments and Suggestions for Authors

Dear authors,

1.General comments

This study is a cross-sectional study that evaluated the efficacy of two self-report scales, MAST and BDI-SF, which are simple and cost-effective, in order to differentiate between SI (suicidal ideation only) and SA (a history of suicide attempts) in outpatients and inpatients of the departments of neurology and psychiatry for children aged 12 to 18 in Italy. The aim of this study was to identify differences in specific items that could distinguish SA, which is particularly at high risk of suicide. As suicide in adolescence is a global health issue, we believe that early detection, prevention, and intervention of suicidal tendencies in this period are extremely important topics. However, there are some points that are unclear in the results and interpretation of the analysis, and there are points that should be improved at this time.

2.Specific comments

a)Major

ⅰ)

The fact that the majority of the subjects of analysis were female is a very important limitation when interpreting the results of this study. I think that appropriate additional analysis (such as adjusting for gender) is necessary in this regard, but there is a possibility that further research could be conducted by narrowing the analysis to only females. In that case, issues specific to women should be given in the introduction and discussion.

ⅱ)

The author states that it is important to note that in SA, when RL scores are high, the BDI-SF suicide item scores are low. However, since the correlation between RL and the BDI-SF is not significant, please explain the basis for this interpretation. RL is an important subscale that produces different results in SI and SA. Even if the results in Table 2 are as expected, the results for RL in Table 4 are very interesting. Furthermore, the fact that the BDI-SF items related to suicidal ideation are no longer significantly related to SI also shows the significance of using these two scales. I would like you to consider this part clearly.

bMinor

ⅰ)

You should add an explanation in the introduction to support the use of the BDI-SF, which is the main scale in this study. Please carefully explain the reasons for using the MAST and BDI-SF in combination, including past studies that have used self-reports.

ⅱ)

The results of the descriptive statistics in lines 168-185 would be easier to understand if they were shown in a table, such as by adding them to Table 1.

ⅲ)

The p-value in Table 2 can be either a number or an asterisk.

ⅳ)

Why did you choose only these three items out of the 30 items in MAST in Table 3? Please add the reason for your choice. Alternatively, since I think that these items are not necessarily essential to the main purpose of this study, it may be better to delete Table 3.

ⅴ)

Please indicate in the footnote the meaning of the number of asterisks in Tables 4 and 5.

ⅵ)

The results of 3.5.3 on line 245 are particularly important, but it was difficult to understand them from the text alone. Why not show the BDI-SF and MAST subscales in a table rather than as text?

ⅷ)

On line 285, you state that the results emphasize that SI has fewer risk factors and more protective factors than SA. Which results give evidence of this?

Author Response

1.General comments

Comment 1: This study is a cross-sectional study that evaluated the efficacy of two self-report scales, MAST and BDI-SF, which are simple and cost-effective, in order to differentiate between SI (suicidal ideation only) and SA (a history of suicide attempts) in outpatients and inpatients of the departments of neurology and psychiatry for children aged 12 to 18 in Italy. The aim of this study was to identify differences in specific items that could distinguish SA, which is particularly at high risk of suicide. As suicide in adolescence is a global health issue, we believe that early detection, prevention, and intervention of suicidal tendencies in this period are extremely important topics. However, there are some points that are unclear in the results and interpretation of the analysis, and there are points that should be improved at this time.

Response 1: We thank the reviewer for taking the time to read the manuscript and making comments.

2.Specific comments

a)Major

ⅰ)

Comment 2: The fact that the majority of the subjects of analysis were female is a very important limitation when interpreting the results of this study. I think that appropriate additional analysis (such as adjusting for gender) is necessary in this regard, but there is a possibility that further research could be conducted by narrowing the analysis to only females. In that case, issues specific to women should be given in the introduction and discussion.

Response 2: We really appreciated the reviewer’s comment, and we agree with the point raised. We plan to rerun the analysis after enlarging the sample size and enrolling a more heterogeneous population to avoid gender bias. We updated the manuscript, specifying that we presented preliminary data and highlighted this limitation as the reviewer suggested.

ⅱ)

Comment 3: The author states that it is important to note that in SA, when RL scores are high, the BDI-SF suicide item scores are low. However, since the correlation between RL and the BDI-SF is not significant, please explain the basis for this interpretation. RL is an important subscale that produces different results in SI and SA. Even if the results in Table 2 are as expected, the results for RL in Table 4 are very interesting. Furthermore, the fact that the BDI-SF items related to suicidal ideation are no longer significantly related to SI also shows the significance of using these two scales. I would like you to consider this part clearly.

Response 3: We thank the reviewer for this comment. To address the suggestion from another reviewer, we removed Table 3 and all the data related to BDI-SF-specific items because it was considered not necessarily essential to the main purpose of the study. However, as you recommended, we deepened the discussion about the results reported in former Table 4, now Table 3, about the not significant results between RL and BDI-SF in SA.

b)Minor

ⅰ)

Comment 4: You should add an explanation in the introduction to support the use of the BDI-SF, which is the main scale in this study. Please carefully explain the reasons for using the MAST and BDI-SF in combination, including past studies that have used self-reports.

Response 4: We followed the reviewer’s suggestion and better discussed, with literature support, why we used those questionnaires.

ⅱ)

Comment 5: The results of the descriptive statistics in lines 168-185 would be easier to understand if they were shown in a table, such as by adding them to Table 1.

Response 5: We thank the reviewer for this suggestion. We moved the data in Table 1 and enlarged it with more data.

ⅲ)

Comment 6: The p-value in Table 2 can be either a number or an asterisk.

Response 6: We harmonised all the Tables to make them more consistent.

ⅳ)

Comment 7: Why did you choose only these three items out of the 30 items in MAST in Table 3? Please add the reason for your choice. Alternatively, since I think that these items are not necessarily essential to the main purpose of this study, it may be better to delete Table 3.

Response 7: We addressed the reviewer’s suggestion of deleting Table 3.

ⅴ)

Comment 8: Please indicate in the footnote the meaning of the number of asterisks in Tables 4 and 5.

Response 8: We thank the reviewer for the comment. We added the required footnote and updated the data in the tables.

ⅵ)

Comment 9: The results of 3.5.3 on line 245 are particularly important, but it was difficult to understand them from the text alone. Why not show the BDI-SF and MAST subscales in a table rather than as text?

Response 9: We addressed the suggestion of adding a table (named Table 4) to make linear regression results easier to read.

ⅷ)

Comment 10: On line 285, you state that the results emphasize that SI has fewer risk factors and more protective factors than SA. Which results give evidence of this?

Response 10: We thank the reviewer for raising this point. We deepened the Discussion section to make it clearer.

Reviewer 3 Report

Comments and Suggestions for Authors

Thank you for the opportunity to review this manuscript examining self-report predictors of suicidal behaviour attempts. Participants were 78, of at least average intelligence, in/outpatient Italian language speaking adolescents. They had been assessed with self-report measures of suicidal ideation, suicidal attempts, depressive symptoms, as well as suicidal protective and risk factors. Two cohorts of young people with suicidal attempts (n = 33) and suicidal ideation only (n = 45, 37 females) were retained for the study. Other comorbidities were also screened for. The study is exploratory in its nature as it attempts to provide clinicians with information as to which scales from commonly used clinical questionnaires can be used by the clinician to assess potential risk of suicidal behaviour/attempts.

As the authors rightly stress, information which assists is early identification and individualisation of treatments for young people presenting with suicidal behaviours is currently a matter of urgency, as such, this is a highly relevant study.

Overall, I felt that the literature review was appropriately succinct. The statistical analysis was well developed and justifiable given the sample’s nature and size. I would suggest that as multiple comparisons are being made some adjustment procedure for the significance levels may be used but this is not always the case (see Rubin, 2021, https://doi.org/10.1007/s11229-021-03276-4) the authors reasoning may be developed in the methodology section.

Some clarifications: when using the C-SSRS to create the groups, did this mean that the SI group had no ‘yes’ responses to any of the suicidal behaviour questions?  

I note that the authors address the most important limitations (sex biased towards females, high degree of pathology in the participants, self-report, etc.) in their discussion. Not mentioned, is the fact that the participants were self-selecting. That is, they agreed to participate in the study from a larger cohort that was approached. Given some personality pathology was present, I believe this is worth mentioning.

Author Response

Comment 1: Thank you for the opportunity to review this manuscript examining self-report predictors of suicidal behaviour attempts. Participants were 78, of at least average intelligence, in/outpatient Italian language speaking adolescents. They had been assessed with self-report measures of suicidal ideation, suicidal attempts, depressive symptoms, as well as suicidal protective and risk factors. Two cohorts of young people with suicidal attempts (n = 33) and suicidal ideation only (n = 45, 37 females) were retained for the study. Other comorbidities were also screened for. The study is exploratory in its nature as it attempts to provide clinicians with information as to which scales from commonly used clinical questionnaires can be used by the clinician to assess potential risk of suicidal behaviour/attempts.

Response 1: We thank the reviewer for taking the time to read the manuscript and making comments.

Comment 2: As the authors rightly stress, information which assists is early identification and individualisation of treatments for young people presenting with suicidal behaviours is currently a matter of urgency, as such, this is a highly relevant study.

Response 2: We appreciated that the reviewer pointed out the strengths of the study and gave positive comments.

Comment 3: Overall, I felt that the literature review was appropriately succinct. The statistical analysis was well developed and justifiable given the sample’s nature and size. I would suggest that as multiple comparisons are being made some adjustment procedure for the significance levels may be used but this is not always the case (see Rubin, 2021, https://doi.org/10.1007/s11229-021-03276-4) the authors reasoning may be developed in the methodology section.

Response 3: Thank you for the suggestion. We did not consider corrections necessary for the present study since we considered it an exploratory study and tried to test every possible hypothesis according to an individual testing approach.

Comment 4: Some clarifications: when using the C-SSRS to create the groups, did this mean that the SI group had no ‘yes’ responses to any of the suicidal behaviour questions? 

Response 4: We thank the reviewer for raising this point. We clarified this in the Participants section.

Comment 5: I note that the authors address the most important limitations (sex biased towards females, high degree of pathology in the participants, self-report, etc.) in their discussion. Not mentioned, is the fact that the participants were self-selecting. That is, they agreed to participate in the study from a larger cohort that was approached. Given some personality pathology was present, I believe this is worth mentioning.

Response 5: We appreciated the suggestion to make the limitation section more precise, and we addressed the reviewer’s request. Moreover, we discussed the presence of personality traits in the Discussion (from about line 300 on).

Reviewer 4 Report

Comments and Suggestions for Authors

The manuscript is well-written and the results' presentation is easy to follow. The only suggestion is that the paper will be strengthened by reviewing and discussing suicide theories and related studies in Introduction. 

Comments on the Quality of English Language

The writing is good.

Author Response

Comment 1: The manuscript is well-written and the results' presentation is easy to follow. The only suggestion is that the paper will be strengthened by reviewing and discussing suicide theories and related studies in Introduction. 

Response 1: We thank the reviewer for the positive comments. We improved the Introduction as suggested.

Comment 2: The writing is good.

Response 2: We appreciate that the manuscript appears easy to read.

Round 2

Reviewer 2 Report

Comments and Suggestions for Authors

I have confirmed the revisions.

You have added a reason for using the two main scales, so the significance of this study has become clearer. I would also like to thank you for adding a discussion of the results of the RL and BDI-SF in the SA. I look forward to seeing your research develop further.

I would like to make a few additional minor comments.

i)

Table 1 shows the characteristics of the subjects, but there is too much information. Is the family history necessary for this paper? You should only include basic attributes, important variables, and other information related to the discussion. In addition, I think that the table would be easier to read if the socioeconomic status, school performance, and social relationships were displayed as binary values such as good or bad.  Furthermore, since it is not clear from just the word “Gender” in Table 1, you should add females, etc.

(ii)

As I mentioned in my previous comment, for the p-values in Tables 1, 2 and 4, either the value or an asterisk is enough.

Author Response

Comments 1: You have added a reason for using the two main scales, so the significance of this study has become clearer. I would also like to thank you for adding a discussion of the results of the RL and BDI-SF in the SA. I look forward to seeing your research develop further.

Response 1: We really appreciate the reviewer’s comments.

Comments 2: Table 1 shows the characteristics of the subjects, but there is too much information. Is the family history necessary for this paper? You should only include basic attributes, important variables, and other information related to the discussion. In addition, I think that the table would be easier to read if the socioeconomic status, school performance, and social relationships were displayed as binary values such as good or bad.  Furthermore, since it is not clear from just the word “Gender” in Table 1, you should add females, etc.

Response 2: We agree and updated Table 1 according to the reviewer’s suggestions. We also removed all the unnecessary information regarding family history. We thank the reviewer for the suggestion to merge some variables and display them in binary values. However, to address a previous review request in which it was requested to clarify the different levels of functioning/performances and SES shown in the degrees of freedom, we chose to retain the non-binary division of sociodemographic variables shown in Table 1.

Comment 3: As I mentioned in my previous comment, for the p-values in Tables 1, 2 and 4, either the value or an asterisk is enough.

Response 3: Thank you for the comment. We removed all the asterisks in Tables 1, 2, and 4 and let the p-values only.